# Transcriptomic Analysis of PDCoV-Infected HIEC-6 Cells and Enrichment Pathways PI3K-Akt and P38 MAPK

**DOI:** 10.3390/v16040579

**Published:** 2024-04-09

**Authors:** Yuhang Jiang, Guoqing Zhang, Letian Li, Maopeng Wang, Jing Chen, Pengfei Hao, Zihan Gao, Jiayi Hao, Chang Li, Ningyi Jin

**Affiliations:** 1College of Veterinary Medicine, Northwest A&F University, Xianyang 712100, China; m13843073620@163.com (Y.J.); haojiayi321@163.com (J.H.); 2Research Unit of Key Technologies for Prevention and Control of Virus Zoonoses, Chinese Academy of Medical Sciences, Changchun Institute of Veterinary Medicine, Chinese Academy of Agricultural Sciences, Changchun 130122, China; zguoqing9908@163.com (G.Z.); letian823@163.com (L.L.); lly5464@163.com (J.C.); hpfbarry@foxmail.com (P.H.); gzh835093861@163.com (Z.G.); 3Wenzhou Key Laboratory for Virology and Immunology, Institute of Virology, Wenzhou University, Chashan University Town, Wenzhou 325000, China; wangmaopenga@126.com

**Keywords:** coronavirus, porcine deltacoronavirus, transcriptome, PI3K-Akt pathway, P38 MAPK pathway

## Abstract

Porcine Deltacoronavirus (PDCoV) is a newly identified coronavirus that causes severe intestinal lesions in piglets. However, the understanding of how PDCoV interacts with human hosts is limited. In this study, we aimed to investigate the interactions between PDCoV and human intestinal cells (HIEC-6) by analyzing the transcriptome at different time points post-infection (12 h, 24 h, 48 h). Differential gene analysis revealed a total of 3560, 5193, and 4147 differentially expressed genes (DEGs) at 12 h, 24 h, and 48 h, respectively. The common genes among the DEGs at all three time points were enriched in biological processes related to cytokine production, extracellular matrix, and cytokine activity. KEGG pathway analysis showed enrichment of genes involved in the p53 signaling pathway, PI3K-Akt signaling pathway, and TNF signaling pathway. Further analysis of highly expressed genes among the DEGs identified significant changes in the expression levels of BUB1, DDIT4, ATF3, GBP2, and IRF1. Comparison of transcriptome data at 24 h with other time points revealed 298 DEGs out of a total of 6276 genes. KEGG analysis of these DEGs showed significant enrichment of pathways related to viral infection, specifically the PI3K-Akt and P38 MAPK pathways. Furthermore, the genes EFNA1 and KITLG, which are associated with viral infection, were found in both enriched pathways, suggesting their potential as therapeutic or preventive targets for PDCoV infection. The enhancement of PDCoV infection in HIEC-6 was observed upon inhibition of the PI3K-Akt and P38 MAPK signaling pathways using sophoridine. Overall, these findings contribute to our understanding of the molecular mechanisms underlying PDCoV infection in HIEC-6 cells and provide insights for developing preventive and therapeutic strategies against PDCoV infection.

## 1. Introduction

PDCoV is a recently discovered coronavirus affecting porcine populations, initially identified in Hong Kong in 2012 [1]. Since the emergence of associated diarrhea in the United States in early 2014 [2], its circulation has been reported in several countries, such as South Korea [3], Thailand [4], Laos [5], Japan [6], and China [7]. Notably, the clinical symptoms caused by PDCoV infection resemble those caused by transmissible gastroenteritis virus (TGEV) and porcine epidemic diarrhea virus (PEDV). The earlier emergence of PDCoV was unnoticed and consequently underestimated the risks posed to newborn piglets. Typical clinical manifestations associated with PDCoV infection include vomiting, severe watery diarrhea, dehydration, and atrophic enteritis [8], along with notable lesions in the stomach and mild lung pathology [9]; causing death in piglets with this infection. Apart from negatively impacting animal welfare, PDCoV significantly affects pig production efficiency. Due to its high infectivity rate, PDCoV outbreaks can rapidly spread through pig farms, leading to significant economic losses in the swine industry.

PDCoV is an enveloped, single-stranded positive-sense RNA virus with a diameter ranging from 60 to 180 nm [10]. Comprising a genome of approximately 25.4 kb [11], PDCoV encodes four structural proteins and four nonstructural proteins, arranged in the following order: 5′ untranslated region (UTR), open reading frames 1a and 1b, S, E, M, NS6, N, [2] NS7, and 3′ UTR. An analysis of sequence similarity and recombination patterns indicates that PDCoV HKU15 (PorCoV HKU15) is closely related to Sparrow Coronavirus (Sp-CoV) HKU17, belonging to the same Delta coronavirus species [1]. Furthermore, the newly discovered Sp-CoV genome (Genbank No. MG812375) shares a higher degree of similarity with PDCoV [12]; PDCoV has demonstrated the ability to infect a diverse range of hosts both in vitro [13,14,15] and in vivo [1,16,17]. The virus has public health significance, as it has been documented to infect children [18]. Although debate exists around the role of the porcine Aminopeptidase N (APN) receptor as a point of invasion for PDCoV [19], it has been confirmed that the receptor-binding domain (RBD) of PDCoV strains recognizes specific residues on both human and porcine APNs, with higher binding affinity to the human form. Remarkably, identical RBM residues with over 96.1% sequence identification of RBD are conserved across different PDCoV strains, enabling cross-species transmission by homologous receptor binding [20]. Hence, the possibility of cross-species transmission of PDCoV between pigs and humans must be considered.

In this study, HIEC-6 cells were subjected to infection with the PDCoV virus. Subsequently, samples were collected at designated time points (12 h, 24 h, and 48 h post-infection) to perform transcriptome sequencing analysis. The obtained data were further subjected to difference analysis, enrichment analysis, and clustering analysis for validation to provide theoretical support for the observed experimental results.

## 2. Materials and Methods

### 2.1. Cells and Antibodies

Swine Testis (ST) and HIEC-6 cells were maintained in Dulbecco’s modified Eagle medium (DMEM) (Gibco, Billings, MT, USA) supplemented with 10% fetal bovine serum (FBS) (Gibco, USA) and a 1% antibiotic antimycotic solution (Solarbio, Beijing, China) at 37 °C in a humidified atmosphere of 5% CO_2_ [21]. ST and HIEC-6 cells were purchased from ATCC. The anti-PDCoV N rabbit monoclonal antibody was prepared by our laboratory. HRP-conjugated goat anti-rabbit IgG (A0208, Beyotime, Shanghai, China) was procured from Beyotime. GoTaq^®^ QpcR Master Mix (A600A, Promega, Madison, WI, USA) was acquired from Promega. 

### 2.2. Virus Culture

PDCoV-MW816149 (GenBank accession no. MW816149) was a kind gift from Dr. Aiqing Jia and was isolated in Guangdong Province in 2018. The PDCoV was cultured using porcine testicular epithelial cells (ST) in this study. For PDCoV infection, the virus was diluted using DMEM containing 10 μg/mL Pancreatin (Solarbio, Beijing, China; T1350) added to the ST cells. Upon reaching 50% cytopathic effect (CPE), both the cells and supernatants were harvested, and cells were freeze–thawed three times. The viral titer of PDCoV was 1 × 10^6^ TCID_50_/mL by the Reed Muench method.

### 2.3. Drug Exposure

Sophoridine was solubilized in dimethyl sulfoxide (DMSO) and subsequently stored. Prior to usage, completely solubilized sophoridine was introduced into the cell culture medium DMEM at a concentration of 0.2 mg/mL. Following a 12 h exposure of HIEC-6 cells to sophoridine, samples were collected for subsequent testing.

### 2.4. One-Step Growth Curve

HIEC-6 cells were infected with PDCoV at a multiplicity of infection (MOI) of 1 at 12, 24, 36, and 48 h. The resulting supernatant was collected and subjected to the evaluation of the 50% tissue culture infectious dose (TCID_50_) using the Reed Muench method. Briefly, HIEC-6 cells were cultured in 96-well plates at a density of 1 × 10^5^ cells/well for 12 h, followed by three washes with phosphate-buffered saline (PBS). The collected supernatant was then diluted 10-fold (ranging from 10^−1^ to 10^−10^) with a cell maintenance solution containing trypsin at a final concentration of 10 mg/mL. Subsequently, the cells were inoculated with the diluted virus and incubated at 37 °C with 5% CO_2_ for 12, 24, 36, and 48 h. The cytopathic effect (CPE) was observed daily using an inverted microscope. The TCID_50_ for each virus was calculated following the method described by Reed and Muench.

### 2.5. Western Blotting

HIEC-6 cells were infected with PDCoV at a multiplicity of infection (MOI) of 1. Cell samples were collected at 12, 24, 36, and 48 h and stored at −80 °C with cell lysis buffer for Western and IP (Beyotime, Shanghai, China) and 1 mM Phenylmethanesulfonyl fluoride (PMSF Beyotime, China). The stored cell samples were subjected to ultrasonic disruption, and the protein concentration in the lysates was evaluated using the BCA protein assay reagent (Beyotime, Shanghai, China). The lysates were further denatured by incubating in a sample buffer (Phygene, Fuzhou, China) at 100 °C for 10 min. Protein gel electrophoresis was performed using a 12.5% denaturing gel, followed by transfer onto nitrocellulose membranes (Millipore, MA, USA). The membranes were blocked with 5% skim milk-TBST solution for 1 h at room temperature and then incubated with primary antibody overnight at 4 °C. After washing the membranes with TBS-T three times, they were incubated with horseradish peroxidase (HRP)-conjugated secondary antibody for 30 min at room temperature. Following further washes with TBS-T three times, protein bands were detected using a chemiluminescence detection kit (Pierce Biotechnology, Rockford, IL, USA).

### 2.6. Transmission Electron Microscopy of PDCoV Virus Particles

The viral supernatant was obtained through centrifugation at 8000 rpm for 10 min. The virus was added to a centrifuge tube containing 20% sucrose and subjected to ultracentrifugation at 30,000 rpm for 2 h at 4 °C. Following ultracentrifugation, the virus particles were resuspended and subsequently adsorbed onto a copper mesh. The adsorbed particles were stained with a 1% phosphotungstic acid buffer for 1–2 min, allowed to dry, and finally utilized for observation using transmission electron microscopy.

### 2.7. Sample Preparation and RNA Extraction

HIEC-6 were cultured in T75 cell culture flasks and subsequently infected with PDCoV at an MOI of 1. Mock-infected cells were also included as a control group. Three flasks of cells from each group were used as independent biological replicates, resulting in a total of 12 samples. At 12, 24, and 48 h post-infection, the samples were collected and lysed using TRIzol reagent (Beyotime, Shanghai, China). The collected samples were then stored in liquid nitrogen and subsequently transferred to Beijing Novogene Company for transcriptome sequencing.

### 2.8. RNA-Seq Analysis

To construct a cDNA library, total RNA was used as the starting material. The mRNA molecules with polyA tails were enriched using Oligo (DT) magnetic beads. The initial quantification of the library was performed using a Qubit2.0 Fluorometer, and the library was diluted to a concentration of 1.5 ng/μL. Subsequently, the insert size of the library was assessed using an Agilent 2100 bioanalyzer. Once the insert size met the expected criteria, the library’s effective concentration was quantified using RT-qPCR to ensure a concentration higher than 2 nM, thus guaranteeing the quality of the library. Comparative analysis was carried out using the Illumina NovaSeq 6000 platform.

### 2.9. Differential Gene Analysis

Differential expression analysis was conducted on two conditions or groups, each with two biological replicates, utilizing the DESeq2 R package (version 1.20.0). DESeq2 offers statistical algorithms for assessing differential expression in digital gene expression data by employing a model based on the negative binomial distribution. The resulting *p*-values were subjected to adjustment using Benjamini and Hochber’s method to control the false discovery rate. A significance threshold was set, whereby values less than or equal to 0.05 and absolute log2 (foldchange) values greater than or equal to 1.5 were considered indicative of significant differential expression.

### 2.10. GO and KEGG Enrichment Analysis

The differential expression genes were subjected to Gene Ontology (GO) enrichment analysis using clusterProfiler (3.8.1) software, which accounted for gene length bias correction. GO terms with a corrected *p*-value less than 0.05 were considered significantly enriched by the DEGs. Genes and Genomes (KEGG) is a database resource that provides insights into the higher-level functions and utilities of biological systems, such as cells, organisms, and ecosystems, based on molecular-level information, particularly from large-scale molecular datasets generated by genome sequencing and other high-throughput databases. We performed statistical enrichment analysis of DEGs in KEGG pathways using clusterProfiler (3.8.1) software.

### 2.11. RT-qPCR Validation

Total RNA was extracted from cells (Sangon Biotech, Shanghai, China) according to the manufacturer’s instructions. cDNA was analyzed using the Fast Start Universal SYBR Green Master Mix (Roche, San Francisco, CA, USA) for quantitative PCR (qPCR). We selected glyceraldehyde-3-phosphate dehydrogenase (GAPDH) as the reference gene and determined the relative gene expression levels of several target candidate genes through RT-qPCR. To assess the changes in DEGs in PDCoV-infected HIEC-6, primers were designed for each gene based on the full-length genomic sequences of each gene discovered in GenBank using the (NCBI Primer-BLAST) software (5.0) (Table 1).

### 2.12. Statistical Analysis

Statistical significance between groups was determined using GraphPad Prism, version 9.3.1. Data were presented as means ± standard errors of the means (SEMs) in all experiments and analyzed using a one-way ANOVA and *t*-test, and a *p*-value of <0.05 was considered to be statistically significant.

## 3. Results

### 3.1. Validation of PDCoV Virus Infection of Different Cells

To ensure the subsequent experiments, a large-scale virus expansion was conducted using the PDCoV-sensitive cell line ST cells (Figure 1A). Cell morphology changes were observed after 24 h of virus infection, including cell crumpling and floating (Figure 1B). The expression level of PDCoV N protein was detected using Western blotting, confirming successful viral proliferation in the cells (Figure 1C). Transmission electron microscopy analysis further revealed the presence of crown-type viral particles with spines in the viral supernatant (Figure 1D). These results demonstrated the successful amplification of PDCoV in ST cells, with an infection titer of 10^6^/0.1 mL TCID_50_ as determined by tissue culture infectious dose 50 (TCID_50_) and a phage plaque assay.

Subsequently, HIEC-6 cells were infected with PDCoV at an MOI of one (Figure 2A). The expression of PDCoV N protein was detected at different time points (12, 24, 36, and 48 h) post-infection using protein blotting. N protein expression was detected at all time points and increased with the duration of infection, indicating the successful infection of HIEC-6 cells by PDCoV (Figure 2B). The results were further confirmed using a quantitative polymerase chain reaction (qPCR) and viral titer analysis; the results obtained from the study suggest a persistent elevation in the expression levels of the PDCoV N gene (Figure 2C), showing that PDCoV reached a TCID50 of more than 10^5.8^/0.1 mL at 24 h post-infection, with minimal changes observed at 36 and 48 h (Figure 2D).

### 3.2. Evaluation of Transcriptome Sequencing Data

We designed transcriptome experiments (Figure 3) with different infection cycles of PDCoV-infected HIEC-6 cells. In total, 6.25 GB of high-quality transcriptome sequencing data per sample was generated using the Illumina NovaSeq 6000 platform. Stringent quality control measures were applied to ensure the suitability of the data for subsequent expression level analysis. The Q30 percentage, which reflects data accuracy, exceeded 90.4% for all samples. The GC content of the clean data ranged from 52 to 55.32% (Table 2). The clean reads of superior quality were aligned to the Bos Taurus reference genome (CASAVA) for further analysis. Approximately 94.14% to 95.93% of the clean reads were successfully mapped to the reference genome, with 91.51% to 93.63% of the clean reads being uniquely mapped.

### 3.3. Transcriptome of PDCoV-Infected HIEC-6

The transcriptome encompasses a total of 23,085 annotated genes at 12 h, 22,625 annotated genes at 24 h, and 23,620 annotated genes at 48 h. Within these time points, 800, 1274, and 1047 genes, respectively, displayed differential expression. Specifically, there were 362 genes up-regulated at 12 h, 353 genes up-regulated at 24 h, and 793 genes up-regulated at 48 h. Additionally, there were 360 genes up-regulated at 12 h, 1274 genes up-regulated at 24 h, and 1047 genes up-regulated at 48 h. Conversely, there were 438 genes down-regulated at 12 h, 921 genes down-regulated at 24 h, and 254 genes down-regulated at 48 h (Figure 4A, Appendix A). Volcano plots were generated based on *p*-values below 0.05 and Log2 fold changes exceeding 1.5. The number of DEGs was higher at 24 h compared to 12 h and 48 h (Figure 4B). A Venn diagram revealed a total of 146 shared genes across different time points, consisting of 110 up-regulated genes and 36 down-regulated genes (Figure 4C).

### 3.4. GO and KEGG Analyses of Shared DEGs

A total of 2425 GO enrichment analyses resulted in the identification of 2007 biological processes (BPs), 177 cellular components (CCs), and 240 molecular functions (MFs) (Figure 4D, Appendix A). Enrichment analysis revealed that they were associated with various biological processes, including the positive regulation of collagen metabolic processes, positive regulation of cytokine production, regulation of endothelial cell development, and regulation of cell cycle phase transitions. These processes involve important proteins such as the extracellular matrix, the connexin complex, and the apical junction complex. Furthermore, molecular functions were related to cytokine receptor binding, DNA-directed DNA polymerase activity, and DNA polymerase activity.

Additionally, KEGG analysis demonstrated that pathways were significantly associated with immune responses, such as the IL-17 signaling pathway, p53 signaling pathway, PI3K-Akt signaling pathway, inflammatory bowel disease, TNF signaling pathway, and ECM–receptor interactions (Figure 4E, Appendix A).

### 3.5. RT-qPCR Validation

We selected five genes (ATF3, BUB1, GBP2, IRF1, and DDIT4) with a Log2 fold change > two in the histological data (Figure 5A) to validate the transcriptome data using RT-qPCR. Among these genes, ATF3 showed a continuous increase in gene expression, while BUB1, GBP2, and IRF1 showed a continuous decrease. DDIT4 initially showed low expression, followed by high expression (Figure 5A, Appendix A). FPKM was used to demonstrate the changes in gene expression levels. The qPCR results for ATF3 were consistent with the FPKM trend, showing an increasing trend (Figure 5B). Similarly, the qPCR results for BUB1, GBP2, and IRF1 were consistent with the FPKM trend, showing a downward trend (Figure 5C–E). The qPCR results for DDIT4 were also consistent with the FPKM trend, showing a downward and then upward trend (Figure 5F). Overall, the RT-qPCR results were consistent with the transcriptomics results.

### 3.6. Comparison Transcriptome Data with the Published Article

The heatmap analysis of these DEGs demonstrated that the majority of them exhibited a decline in expression following PDCoV infection (Figure 6C, Appendix A). Additionally, the GO enrichment analysis indicated a significant association of the DEGs with chromosome function (Figure 6D, Appendix A). These findings from the data comparison allowed for a more precise determination of the repertoire of cellular response molecules following viral infection. Furthermore, they supported the reliability of the histological data and underscored the continuous decrease in gene expression subsequent to PDCoV infection.

### 3.7. KEGG Enrichment of DEGs and Heatmap Analysis of Key Pathway DEGs

KEGG analysis was conducted on the 298 DEGs, and the top 20 results are presented in Figure 7A (Appendix A). The KEGG analysis results demonstrated significant enrichment of DEGs in various signaling pathways, including the PI3K-Akt signaling pathway, human papillomavirus infection, P38 MAPK signaling pathway, NOD-like receptor signaling pathway, cell cycle, and ECM–receptor interactions. Among these pathways, the PI3K-Akt signaling pathway and P38 MAPK signaling pathway were identified as the main pathways associated with viral infection. Further analysis of the genes enriched in these two pathways (Figure 7B, Appendix A) revealed the presence of KITLG and EFNA1 in both pathways, suggesting a close association between these pathways and PDCoV infection. These findings highlight the potential of KITLG and EFNA1 as therapeutic or preventive targets for PDCoV infection. The mRNA expression level of the PDCoV N protein exhibited a significant difference in its inhibition of PI3K when treated with sophoridine compared to the vector (Figure 7C). Similarly, the mRNA levels of PI3K and P38 were found to differ in the presence of sophoridine compared to the DMSO control (Figure 7D,E). Additionally, Western blot analysis revealed that the inhibition of both PI3K and P38 resulted in increased PDCoV virus infection (Figure 7F). These findings suggest that both the PI3K and P38 pathways may serve as crucial mechanisms for inhibiting PDCoV infection in HIEC-6 cells.

## 4. Discussion

The outbreak of PDCoV initially emerged in pig populations in the United States in 2014, subsequently spreading to several Asian countries [1]. Clinical manifestations commonly observed in PDCoV infection encompass emesis, profound watery diarrhea, dehydration, atrophic enteritis, as well as notable gastric and mild lung lesions, ultimately leading to mortality in affected piglets. Beyond the deleterious effects on animal well-being, PDCoV exerts a substantial influence on porcine productivity. Given its elevated transmission rate, PDCoV outbreaks can swiftly disseminate within swine facilities, resulting in substantial economic detriments to the pig farming sector. Furthermore, PDCoV exhibits an exceptionally wide range of susceptible hosts. Previous studies have demonstrated the susceptibility of chicken fibroblast DF1 [16,23], HEK-293 [24], and ST [10] cells to PDCoV infection. Notably, in November 2021, the first documented case of PDCoV transmission from pigs to children was reported in Haiti [22]. Currently, human cases of PDCoV infection exhibit only mild symptoms, but it remains uncertain whether PDCoV can be efficiently transmitted to humans. However, if PDCoV undergoes mutations that enhance its infectivity, it could potentially pose a significant challenge to disease control measures.

Several researchers have conducted transcriptomic investigations on PDCoV-infected ST cells and PK-15 cells [25,26,27], elucidating the association between PDCoV infection and innate immunity in these cell types. In a study conducted by Cruz-Pulido et al., a comparison was made between the transcriptomes of PDCoV-infected HIEC-6 and IPEC-J2 cells, revealing striking similarities in the infection pathways [22]. In the present work, we compared our sequencing results with the 24 h PDCoV-infected HIEC-6 cells data from Cruz-Pulido et al. However, due to the differential sequencing methodology employed, our study yielded a higher overall factor than that reported by Cruz-Pulido et al. We present evidence demonstrating the infection of HIEC-6 cells by PDCoV, resulting in morphological changes such as cell shrinkage, detachment, and cell death. While previous studies have established the infectivity of PDCoV in HIEC-6 cells [22], the underlying mechanisms of infection remain poorly understood. To address this gap, we conducted a transcriptomic analysis of PDCoV infection in HIEC-6 cells at various time points. Through this analysis, we identified DEGs and investigated their involvement in pathways associated with viral infection. Our findings revealed a significant enrichment of DEGs in the P38 MAPK and PI3K-Akt pathways. Previous research has suggested a potential link between the antiviral activity of sophoridine and its ability to inhibit the activation of cellular PI3K-Akt and P38 MAPK pathways during herpesvirus infection [28]. These observations, coupled with existing knowledge on the role of PI3K-Akt and P38 MAPK signaling pathways in viral infections, underscore the importance of targeting these pathways for a reduction in viral infections [29]. In the present experimental study, the inhibitory effects of sophoridine on the PI3K and P38 MAPK signaling pathways were investigated in relation to Porcine Deltacoronavirus (PDCoV) infection in HIEC-6 cells. Interestingly, the inhibition of both pathways resulted in a significant increase in PDCoV infection in HIEC-6 cells within a time frame of 12 h post-inhibition. Our findings indicate a significant enrichment of DEGs in the P38 MAPK and PI3K-Akt pathways following PDCoV infection of HIEC-6 cells, suggesting a potential association between PDCoV infection and these pathways, as well as the involvement of KITLG and EFNA1 genes. Notably, the genes KITLG and EFNA1 are found in both the PI3K-Akt and P38 MAPK signaling pathways, suggesting that they may serve as common factors connecting these two pathways. CRNG disruption and overexpression, in conjunction with KITLG, have been observed to interact and engage in viral and pathogenic infections in PK-15 cells [30]. Furthermore, it has been observed that the down-regulation of EFNA1 expression exerts inhibitory effects on the three-dimensional growth of HT29 colon cancer cells [31]. Previous studies have also reported that exhibits a therapeutic effect in experimental colitis by inhibiting the CEBPB/PCK1 and CEBPB/EFNA1 pathways, which are associated with the PI3K-Akt and P38 MAPK pathways [32]. CRNG disruption and overexpression, in conjunction with KITLG, have been observed to interact and engage in viral and pathogenic infections in PK-15 cells [30]. The identification of DEGs in the PI3K-Akt and P38 MAPK pathways, particularly KITLG and EFNA1, suggests their potential significance in the context of PDCoV infection. Our findings shed light on the molecular underpinnings of PDCoV infection in HIEC-6 cells, presenting opportunities for the development of preventive and therapeutic strategies against PDCoV infection.

## 5. Conclusions

The potential risk of widespread infection resulting from PDCoV mutations underscores the urgent need for a comprehensive understanding of the mechanisms underlying PDCoV infection in humans. In this study, we performed transcriptome sequencing analysis of PDCoV infection of HIEC-6 at 12, 24, and 48 h. Five of the identified genes were subjected to validation and comparison with prior studies, wherein it was observed that the PI3K-Akt and P38 MAPK pathways, which have been recognized as antiviral pathways, were significantly enriched among the top 20 pathways. Notably, two DEGs, namely KITLG and EFNA1, were found to be associated with these pathways. Consequently, we observed enhanced PDCoV infection in HIEC-6 cells by suppressing the PI3K and P38 MAPK signaling pathways through the administration of sophoridine, thus validating our hypothesis. We postulate that the PI3K-Akt and P38 MAPK pathways may play a crucial role in the progression of PDCoV infection in HIEC-6 cells.

## Figures and Tables

**Figure 1 viruses-16-00579-f001:**
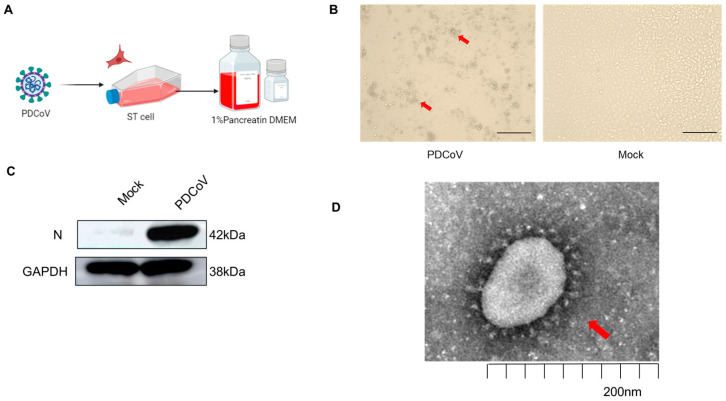
Proliferation of PDCoV in ST cells. (**A**) Process of PDCoV infection in ST cells. (**B**) ST cell lesion caused by PDCoV infection; red arrow positions are cytopathic lesions. The scale bar with 100×.(**C**) PDCoV replication in ST cells. (**D**) Transmission electron microscopy observation of PDCov particles; red arrow positions are PDCoV virus particles. The scale bar corresponds to 200 nm.

**Figure 2 viruses-16-00579-f002:**
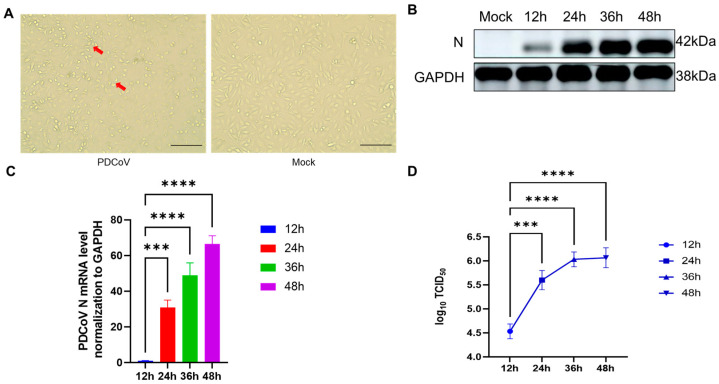
Verification of PDCoV infection in HIEC cells. (**A**) Process of PDCoV infection in HIEC-6 cells; red arrow positions are cytopathic lesions. The scale bar with 100×. (**B**) PDCoV N gene replication in HIEC cells (WB). (**C**) PDCoV N gene replication in HIEC cells (qPCR). (**D**) One-step growth curve of PDCoV in HIEC cells; the analytical approach implemented in this series of analyses involved the utilization of ANOVA (*** *p* < 0.001; **** *p* < 0.0001).

**Figure 3 viruses-16-00579-f003:**
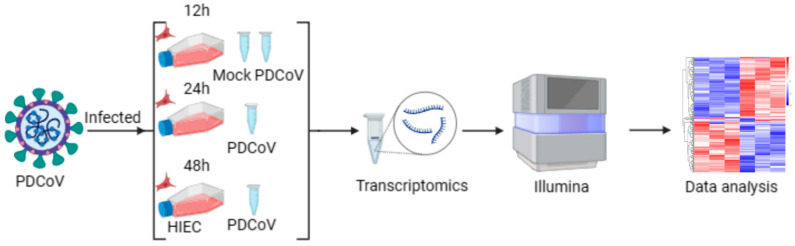
Procedure of transcriptome analysis of PDCoV-infected HIEC-6 cells (mock is HIEC-6 cells collected at 12 h without PDCoV virus infection; PDCoV is HIEC-6 cells collected at 12 h, 24 h, and 48 h, respectively, after PDCoV infection).

**Figure 4 viruses-16-00579-f004:**
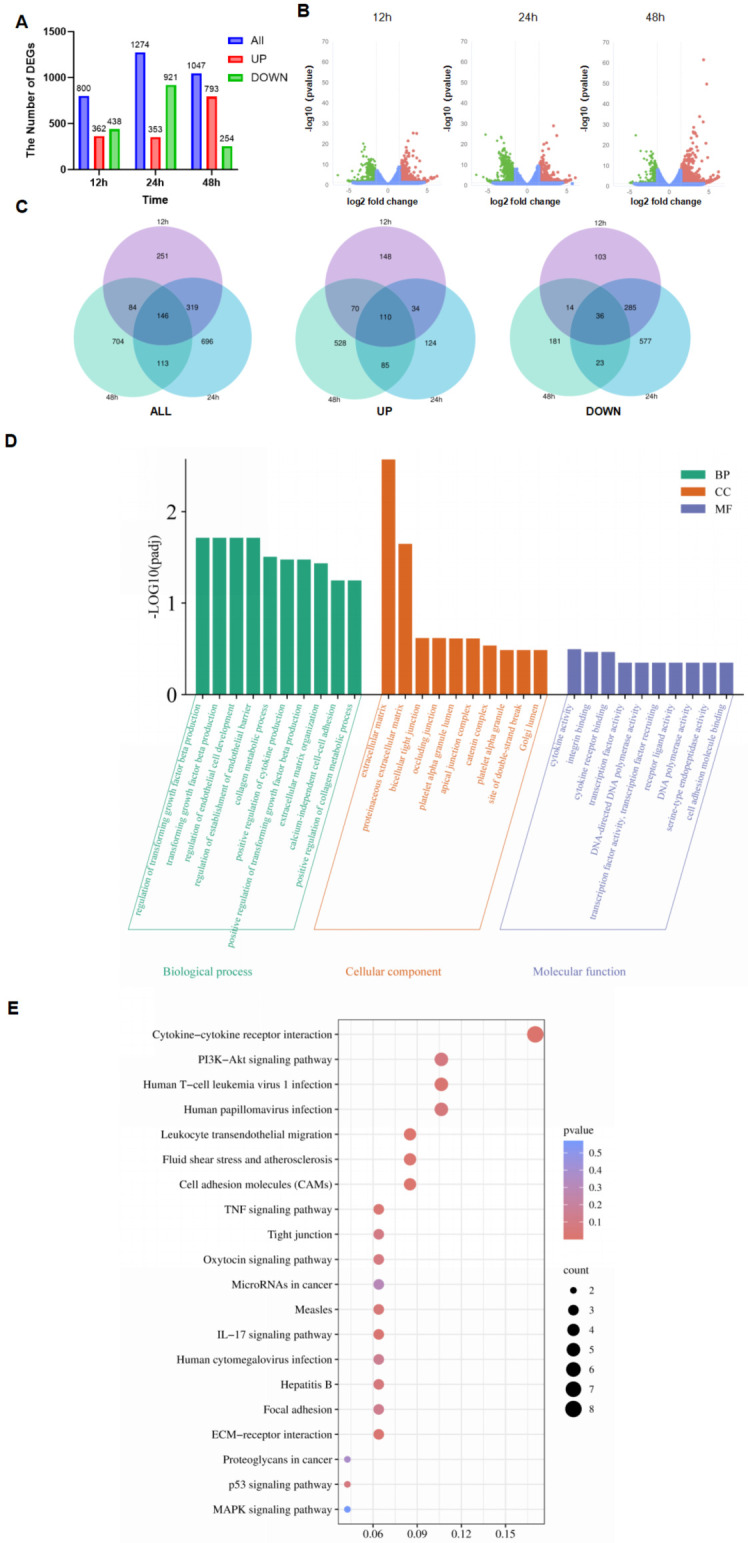
The differentially expressed genes in all transcriptome groups. (**A**) The number of DEGs at different times. (**B**) Volcano map of differential genes changing at different times. (**C**) Comparison of up-regulated genes, down-regulated genes, and all genes at different times with Venn map. Venn (**D**) GO annotation of shared DEGs. (**E**) KEGG annotation of shared DEGs (biological process—BP; cellular components—CCs; molecular function—MF).

**Figure 5 viruses-16-00579-f005:**
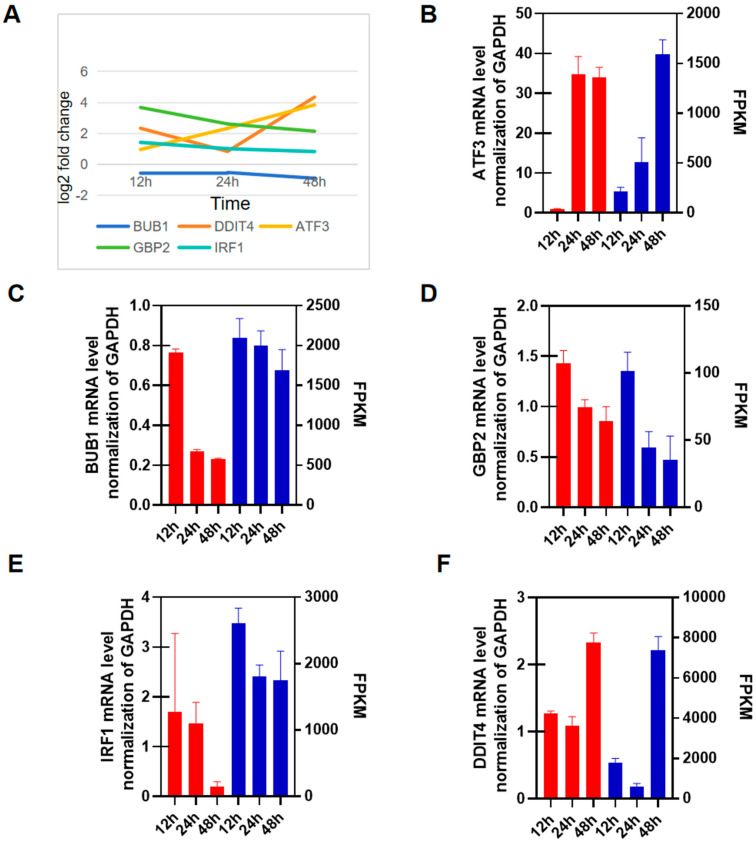
Differentially expressed key genes of Top20. HIEC cells were infected with 1 MOI of PDCoV for 12 h, 24 h, and 48 h. The expression levels of (**A**) FPKM change level of genes (**B**) ATF3, (**C**) BUB1, (**D**) GBP2, (**E**) IRF1, (**F**) and DDIT4 were compared at 12 h with RT-PCR. GAPDH was used as the internal control. The mRNA expression level is represented by red, while the FPKM expression level is represented by blue. FPKM stands for fragments per kilobase of exon model per million mapped fragments.

**Figure 6 viruses-16-00579-f006:**
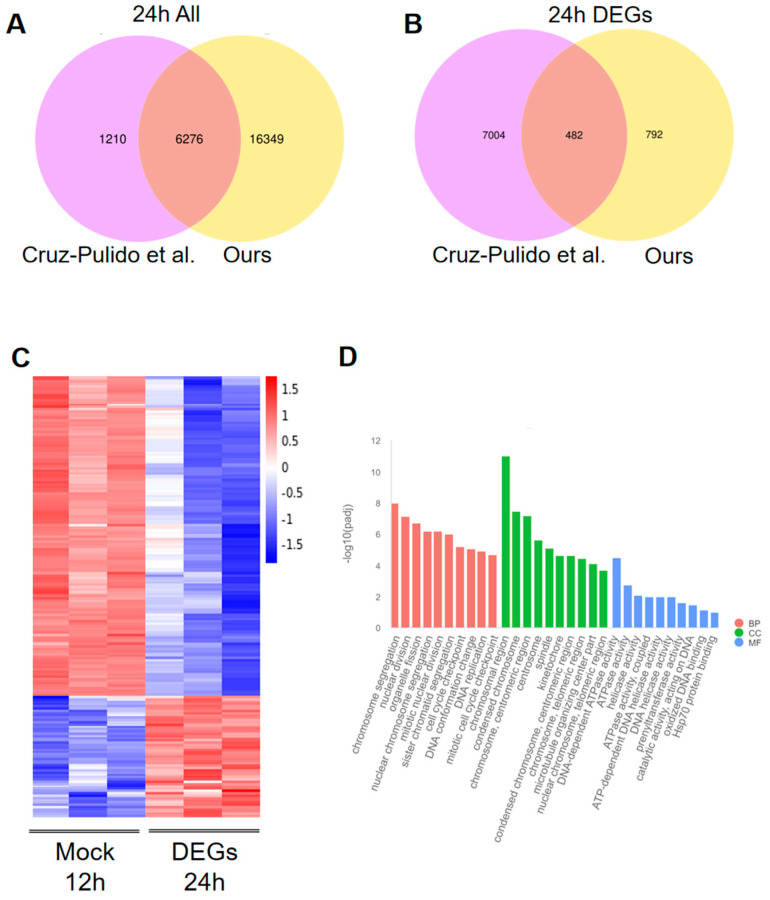
Comparison with 24 h PDCoV-infected HIEC data in the article. (**A**) Venn diagrams of the data in the article compared with all the 24 h genes in this experiment [22]. (**B**) Venn diagrams of data in the article versus this experiment with 24 h DEGs gene comparison. (**C**) Heat map analysis of 482 DEGs. (**D**) GO enrichment of 482 DEGs.

**Figure 7 viruses-16-00579-f007:**
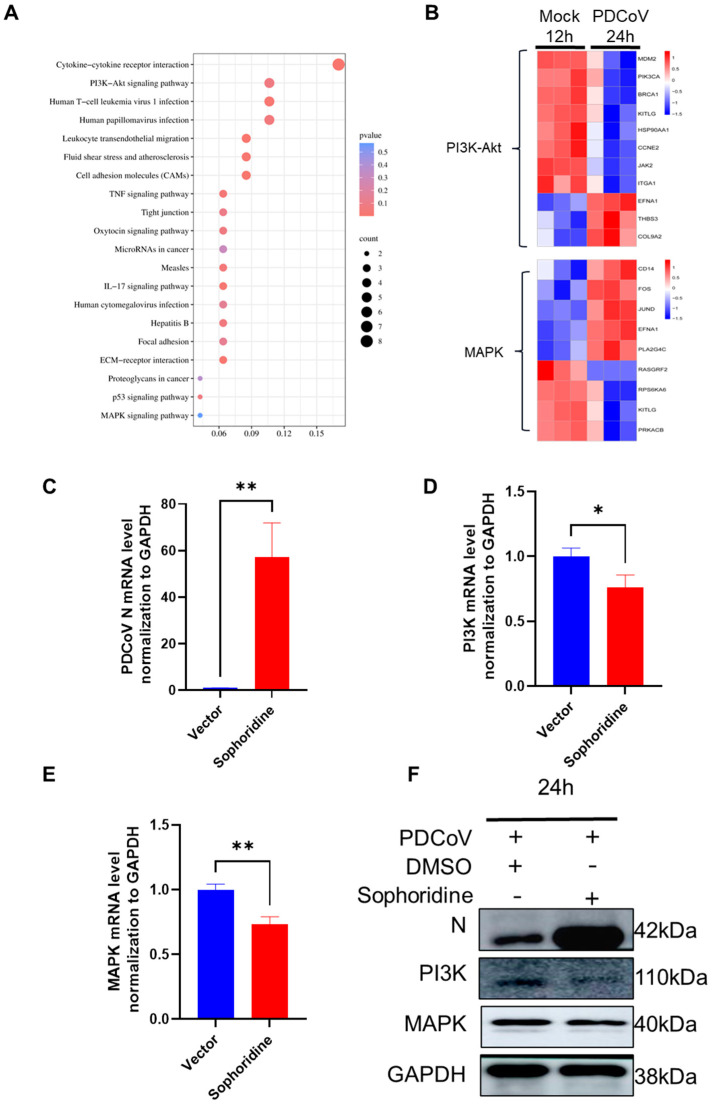
Protein and qPCR analysis of HIEC-6 cells infected with PDCoV after inhibition of the PI3K-Akt and P38 MAPK pathway using sophoridine. (**A**) KEGG analysis of 482 DEGs (the probability of a difference between samples due to sampling error is less than 0.05 or 0.01. Padj: p.adjust). (**B**) Heatmap analysis of enriched genes for PI3K-Akt and P38 MAPK. (**C**) PDCoV N gene replication in HIEC-6 cells (qPCR). (**D**) PI3K gene replication in HIEC-6 cells (qPCR). (**E**) P38 gene replication in HIEC-6 cells (qPCR). (**F**) PDCoV N gene replication in HIEC-6 cells (WB). The analytical approach implemented in this series of analyses involved the utilization of a *t*-test. (* *p* < 0.05; ** *p* < 0.01) (vector is HIEC-6 cells with only DMSO added; sophoridine is HIEC-6 cells with sophoridine added).

**Table 1 viruses-16-00579-t001:** Primers for RT-qPCR in this study.

Name	Sequence
PDCoV-N-F	CTATGAGCCACCCACCAA
PDCoV-N-R	TCCCACTCCCAATCCTGT
*BUB1*-F	GTTGCATCAGGTGGTGGAGA
*BUB1*-R	GAGGTGCCTCTCTTGGGTTC
*DDIT4*-F	GCTTACCTGGATGGGGTGTC
*DDIT4*-R	GCATCAGGTTGGCACACAAG
*ATF3*-F	CTGTCTCGAGACCATGATGCTTCAACATCCA
*ATF3*-R	CTGTCCCGGGTTAGCTCTGCAATGTTCC
*GBP2*-F	ATGAACAAGCTGGCTGGGAA
*GBP2*-R	TCTTGGGATGAGGCACACAC
*IRF1*-F	ACCCTGGCTAGAGATGCAGA
*IRF1*-R	TGCTTTGTATCGGCCTGTGT
*GAPDH*-F	CTACATGGTTTACATGTTCC
*GAPDH*-R	GGATCTCGCTCCTGGAAGAT
qPI3K-F	CTGCAGTTCAACAGCCACAC
qPI3K-R	ACAGGTCAATGGCTGCATCA
qMAPK-F	AGGCTGTTCCCAAATGCTGA
qMAPK-R	CAGATATGGGTGGGCCAGAG

**Table 2 viruses-16-00579-t002:** Summary statistics for sequence quality control and mapped data of samples.

Sample	Raw Reads	Raw Bases	Clean Reads	Clean Bases	Q30	GCpct	Total Map	Multimap	Unique Map
Mock-1	45142592	6.77 G	44517704	6.68 G	90.83	52.52	42706945	1024389	41682556
Mock-2	46280682	6.94 G	45076696	6.76 G	91.35	52.98	43027266	1136101	41891165
Mock-3	44625196	6.69 G	43355404	6.5 G	91.32	52	41228385	1115310	40113075
PDCoV-12h-1	44691544	6.7 G	43856336	6.58 G	90.88	53.47	41847210	1018571	40828639
PDCoV-12h-2	45824392	6.87 G	44782282	6.72 G	90.4	55.32	42330576	1111281	41219295
PDCoV-12h-3	45255846	6.79 G	44379500	6.66 G	90.68	53.71	42067900	1167913	40899987
PDCoV-24h-1	42819744	6.42 G	41660400	6.25 G	91.71	55.48	39508414	1080319	38428095
PDCoV-24h-2	44243262	6.64 G	42104606	6.32 G	91.31	53.55	39879439	1055958	38823481
PDCoV-24h-3	45048018	6.76 G	43430312	6.51 G	91.04	54.52	40885945	1143274	39742671
PDCoV-48h-1	46048916	6.91 G	43818106	6.57 G	90.29	54.21	41277870	1128930	40148940
PDCoV-24h-2	45317606	6.8 G	42285954	6.34 G	90.57	53.06	39831492	1093925	38737567
PDCoV-24h-3	44781216	6.72 G	42845208	6.43 G	90.56	50.91	40670083	1013339	39656744

## Data Availability

The datasets provided in this study can be found in online repositories. The names and access numbers of the repositories/repositories are as follows: the National Center for Biotechnology Information, BioProjects Database, registration number PRJNA1047683 https://www.ncbi.nlm.nih.gov/bioproject/PRJNA1047683 (accessed on 1 December 2023).

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
