# Peer review of "Transcriptomic Analysis of PDCoV-Infected HIEC-6 Cells and Enrichment Pathways PI3K-Akt and P38 MAPK"

_viruses, 2024, doi:10.3390/v16040579_

Round 1

Reviewer 1 Report (Previous Reviewer 1)

Comments and Suggestions for Authors

Comments on the Quality of English Language

Author Response

Point-by-point responses to the comments of the Editor and reviewers

Reviewer 1

Comments to the Author (blind)

Overall, this is a very well executed manuscript with clear concise language and few typographical, grammar or spelling errors.  I commend you on that regard, making this an easy read.  The project is defined very well.  I still have issues with some of the figures lacking appropriate labels, descriptions or being too small and not readable.  The methods for qRT-PCR are still inadequate and need to be expanded.  The green highlights are the problems I found previously that appeared to be properly addressed. You have only addressed about half of the issues I flagged originally and the references in the Introduction that I asked you to add have been misrepresented.

Response: Thank you for the positive comment and the constructive suggestions for improving the manuscript. We have revised our manuscript according to all the issues listed below.

Introduction:

Q1:Line 58-62: you attempted to correct what I flagged you for the first, but your references are still in error.

Reference 13 is in chickensand is in ovo and in vivo.

Reference 14 is in bovine cells

Reference 15 is also in calves but is in vivo.

Reference 16 is not in pigs.  It is in chicken and turkey poults and is in vivo.

Reference 17 is all in vitro and included numerous cell types. 

You should carefully examine you references before inserting them because you got it so wrong here.

The following sentence (62-63) should be removed because it seems to duplicate this sentence. 

Response: Thank you for your very professional advice, and we now make the following changes " PDCoV has demonstrated the ability to infect a diverse range of hosts both in vitro[13-15] and in vivo[1, 16, 17].."at Line 58-59.

References added in the revised manuscript:

  1. Woo, P. C., S. K. Lau, C. S. Lam, C. C. Lau, A. K. Tsang, J. H. Lau, R. Bai, J. L. Teng, C. C. Tsang, M. Wang, B. J. Zheng, K. H. Chan, and K. Y. Yuen. "Discovery of Seven Novel Mammalian and Avian Coronaviruses in the Genus Deltacoronavirus Supports Bat Coronaviruses as the Gene Source of Alphacoronavirus and Betacoronavirus and Avian Coronaviruses as the Gene Source of Gammacoronavirus and Deltacoronavirus." J Virol 86, no. 7 (2012): 3995-4008.
  2. Li, W., R. J. G. Hulswit, S. P. Kenney, I. Widjaja, K. Jung, M. A. Alhamo, B. van Dieren, F. J. M. van Kuppeveld, L. J. Saif, and B. J. Bosch. "Broad Receptor Engagement of an Emerging Global Coronavirus May Potentiate Its Diverse Cross-Species Transmissibility." Proc Natl Acad Sci U S A 115, no. 22 (2018): E5135-e43.
  3. Jung, K., M. Vasquez-Lee, and L. J. Saif. "Replicative Capacity of Porcine Deltacoronavirus and Porcine Epidemic Diarrhea Virus in Primary Bovine Mesenchymal Cells." Vet Microbiol 244 (2020): 108660.
  4. Boley, P. A., M. A. Alhamo, G. Lossie, K. K. Yadav, M. Vasquez-Lee, L. J. Saif, and S. P. Kenney. "Porcine Deltacoronavirus Infection and Transmission in Poultry, United States(1)." Emerg Infect Dis 26, no. 2 (2020): 255-65.
  5. Liang, Q., H. Zhang, B. Li, Q. Ding, Y. Wang, W. Gao, D. Guo, Z. Wei, and H. Hu. "Susceptibility of Chickens to Porcine Deltacoronavirus Infection." Viruses 11, no. 6 (2019).
  6. Jung, K., H. Hu, and L. J. Saif. "Calves Are Susceptible to Infection with the Newly Emerged Porcine Deltacoronavirus, but Not with the Swine Enteric Alphacoronavirus, Porcine Epidemic Diarrhea Virus." Arch Virol 162, no. 8 (2017): 2357-62.

Methods:

Q1:Line 73: Recommended media for HIEC-6 cells is: s OptiMEM 1 Reduced Serum Medium (Gibco Catalog No. 31985) supplemented with: 20 mM HEPES, 10 mM GlutaMAX (Gibco Catalog No. 35050), 10 ng/mL Epidermal Growth Factor (EGF), fetal bovine serum (FBS) to a final concentration of 4%.  Is this what you used?

Response: Thank you very much for your recommendation. We would like to clarify that the methodology mentioned in your query was not utilized in our study, and we have previously addressed this matter in our initial response.

Q3:Line 135: qRT-PCR.

Section 2.10: Did you optimize your primers before using in qRT-PCR? This description of methods for qRT-PCR is inadequate.  Did you determine your primer efficiencies before using?

Response: We appreciate the reviewer’s comments and suggestions. We didn’t optimize primers. Some of them are from references. But it’s a good advice. We will pay attention to this in future experiments. However, we determined the primer efficiencies before use by repeat experiments. Additionally, we rewrote the qRT-PCR section in the revised manuscript as follows: “Total RNA was extracted from cells (Sangon Biotech, Shanghai, China) according to the manufacturer’s instructions. cDNA was analyzed using the Fast Start Universal SYBR Green Master Mix (Roche, San Francisco, CA, USA) for quantitative PCR (qPCR). We selected glyceraldehyde-3-phosphate dehydrogenase (GAPDH) as the reference gene and determined the relative gene expression levels of several target candidate genes through RT-qPCR. The specific primers are shown in Table 1.”

Discussion:

Figures: 

Q1:Figure  1A: work flow shows the use of Pancreatin but there is no mention of this in your methods.

Response: Thank you very much for your careful reviews. We have previously addressed this matter in our previous response to the query on Methods (lines 92-95) sections.

Q3:Figure 1D: Could be lighter.

Response: Thank you very much for your recommendation. We have previously addressed this matter in our initial response.

Q4:Figure 2B:  What are the units represented on the y-axis?  Also, elaborate on what the comparisons are in this figure? Are they all individually compared to Mock or to 12 hours?  Why did you not use ANOVA to see if the differences are significant at all time points between groups? What is WB in the caption?

Response: Thank you very much for your recommendation.These data were all compared to the 12h Mock as a Vector, and we took your suggestion to analyze the data using ANOVA and added the methodology at Line179-182.And WB we have deleted.

Q5:Figure 2C:  It is preferable to log10 your TCID50 results when graphing.  Did you do any statistics on these results?

Response: Thank you very much for your recommendation. We have previously addressed this matter in our initial response.

Q6:The Statistics information should be at the end of the caption. Does it describe D as well?  You should be using ANOVA, I already said this previously.  Please explain why you used the student’s t-test for so many groups.

Response: We appreciate the reviewer’s comments and suggestions. We put Statistics information should be at the end of the caption , Following your recommendation, we employed Analysis of Variance (ANOVA) for all subsequent analytical procedures.

Q7:Figure 4B: You have these volcano plots lined up together so they should all have the same scale on the y-axis for visual comparison of any differences. Also, the label on the x-axis should be log2 Fold Change.  Everything has run together. 

Response: Thank you very much for your recommendation. We have previously addressed this matter in our initial response. The labeling log 2 fold change was separately incorporated on the y-axis.

Q8:Figure 5A: Axes are not properly labeled

Response: We appreciate your response; however, there seems to be a miscommunication.  In our study, the x-axis is denoted as "Time" and the y-axis is labeled as "log 2 fold change." Therefore, it is unclear where the issue of missing labeling arises.

Q9:Any statistics at all here?

Response: Thank you for your response. Unfortunately, we cannot address any statistical concerns as no statistical data has been provided in this context.

Q10:Figures 6A and B are Venn Diagrams not veen

Response: We extend our sincere gratitude for your invaluable suggestion. We would like to reiterate that the aforementioned question has been previously addressed in our initial response

Q11:Figures 7C&E: Normalization is misspelled on the y -axis

Response: Thank you very much for your careful reviews. We have corrected .

Q12:Figure 7 caption.  No statistical information other than p-values some of which are not represented in the figure.

Response: We have accepted your suggestions and made the changes at Line 331-339.Protein and qPCR analysis of HIEC-6 cells infected with PDCov after inhibition of the PI3K-Akt & P38 MAPK pathway using sophoridine, Vector is HIEC-6 cells with only DMSO added, sophoridine is HIEC-6 cells with sophoridine added

Reviewer 2 Report (Previous Reviewer 2)

Comments and Suggestions for Authors

The authors have revised their paper including adding sophoridine treatment to downregulate the PI3K-Akt and P38 MAPK pathways and confirm the effect on viral infection. However they still failed to address the following issue.

In the prior paper published in Viruses in 2021 (cited as ref 17, now ref 22), the detailed transcriptomic analysis of PDCoV-infected human intestinal HIEC-6 cells vs swine IPEC-J2 cells at 24 hr post-infection was described. Although the authors compared their data at 24 hr post-infection of HIEC-6 cells (Fig 6A, 6B) to that in the published paper (now ref 22), they did not address or discuss the large differences seen in Fig 6 A,B between the 2 studies. Also in their new edits and revisions, several are either incorrect or unclearly stated and need to be revised. Someone skilled in English grammar should review the paper.

Comments;

1. l 59-63- the information is incorrect. …”While all the afore-mentioned experiments examined interspecies transmission in various species in vitro, none of them assessed it in vivo.” Statement and Ref are incorrect—in vivo=ref 8,13,15, 16. Ref pigs #14 is for calf cells. L 62-63 is redundant. Also add correct in vitro ref for each spp.

2. L 87-89-  poorly written sentence-rewrite.

3. L321-323 Unclear as written—rewrite.

4. L355-357 missing comparative info from ref 22—these authors compared transcriptomic analysis of PDCoV infection in human (HIEC-6) and pig IPEC-J2 intestinal cells. Also comparisons including ones done on this data for Fig 6 are not discussed.

5. L 360 Ref 22 missing and no discussion of comparative findings

6. L 370-372 poorly written sentence-rewrite. For example: Suppression of the PI3K and P38 MAPK signaling pathways through sophoridine in the HIEC-6 cells augmented PDCoV infection.

7. L374-376 Highlight where the data on KITLG and EFNA1 genes is presented in the results and tie in this data to the discussion. The lengthy discussion of these genes should be germaine to the findings in this study instead of a general discussion. 

Figures:

Fig 1 legend not modified per revised Fig 1—300um marker bar should be 100um and where is this fig?

Fig. 2. A PDCoV  versus mock infection… Quality of Fig 2A is poor and unlikely to reproduce well in the paper.

Fig.3. Legend is Unclear as written—rewrite.

Fig.4, 6. VENN—not veen?

Fig.7. Legend is not clearly written and does not provide info on the treatments (sophoridine, DMSO)

Comments on the Quality of English Language

See concerns with clarity and poorly written sentences in the review.

Author Response

Point-by-point responses to the comments of the Editor and reviewers

Reviewer 2

Comments to the Author (blind)

The authors have revised their paper including adding sophoridine treatment to downregulate the PI3K-Akt and P38 MAPK pathways and confirm the effect on viral infection. However they still failed to address the following issue.

In the prior paper published in Viruses in 2021 (cited as ref 17, now ref 22), the detailed transcriptomic analysis of PDCoV-infected human intestinal HIEC-6 cells vs swine IPEC-J2 cells at 24 hr post-infection was described. Although the authors compared their data at 24 hr post-infection of HIEC-6 cells (Fig 6A, 6B) to that in the published paper (now ref 22), they did not address or discuss the large differences seen in Fig 6 A,B between the 2 studies. Also in their new edits and revisions, several are either incorrect or unclearly stated and need to be revised. Someone skilled in English grammar should review the paper.

Response: Thank you for the positive comment and the constructive suggestions for improving the manuscript. We have revised our manuscript according to all the issues listed below.

Comments;

Q1:l 59-63- the information is incorrect. …”While all the afore-mentioned experiments examined interspecies transmission in various species in vitro, none of them assessed it in vivo.” Statement and Ref are incorrect—in vivo=ref 8,13,15, 16. Ref pigs #14 is for calf cells. L 62-63 is redundant. Also add correct in vitro ref for each spp.

Response: Thank you for your very professional advice, and we now make the following changes " PDCoV has demonstrated the ability to infect a diverse range of hosts both in vitro[13-15] and in vivo[1, 16, 17].."at Line 58-59.

References added in the revised manuscript:

  1. Woo, P. C., S. K. Lau, C. S. Lam, C. C. Lau, A. K. Tsang, J. H. Lau, R. Bai, J. L. Teng, C. C. Tsang, M. Wang, B. J. Zheng, K. H. Chan, and K. Y. Yuen. "Discovery of Seven Novel Mammalian and Avian Coronaviruses in the Genus Deltacoronavirus Supports Bat Coronaviruses as the Gene Source of Alphacoronavirus and Betacoronavirus and Avian Coronaviruses as the Gene Source of Gammacoronavirus and Deltacoronavirus." J Virol 86, no. 7 (2012): 3995-4008.
  2. Li, W., R. J. G. Hulswit, S. P. Kenney, I. Widjaja, K. Jung, M. A. Alhamo, B. van Dieren, F. J. M. van Kuppeveld, L. J. Saif, and B. J. Bosch. "Broad Receptor Engagement of an Emerging Global Coronavirus May Potentiate Its Diverse Cross-Species Transmissibility." Proc Natl Acad Sci U S A 115, no. 22 (2018): E5135-e43.
  3. Jung, K., M. Vasquez-Lee, and L. J. Saif. "Replicative Capacity of Porcine Deltacoronavirus and Porcine Epidemic Diarrhea Virus in Primary Bovine Mesenchymal Cells." Vet Microbiol 244 (2020): 108660.
  4. Boley, P. A., M. A. Alhamo, G. Lossie, K. K. Yadav, M. Vasquez-Lee, L. J. Saif, and S. P. Kenney. "Porcine Deltacoronavirus Infection and Transmission in Poultry, United States(1)." Emerg Infect Dis 26, no. 2 (2020): 255-65.
  5. Liang, Q., H. Zhang, B. Li, Q. Ding, Y. Wang, W. Gao, D. Guo, Z. Wei, and H. Hu. "Susceptibility of Chickens to Porcine Deltacoronavirus Infection." Viruses 11, no. 6 (2019).
  6. Jung, K., H. Hu, and L. J. Saif. "Calves Are Susceptible to Infection with the Newly Emerged Porcine Deltacoronavirus, but Not with the Swine Enteric Alphacoronavirus, Porcine Epidemic Diarrhea Virus." Arch Virol 162, no. 8 (2017): 2357-62.

Q2: L 87-89-  poorly written sentence-rewrite.

Response:Thanks to your suggestions, we've rewritten Line 87-89:”Sophoridine was solubilized in dimethyl sulfoxide (DMSO) and subsequently stored. Prior to usage, completely solubilized sophoridine was introduced into the cell culture medium DMEM at a concentration of 0.2 mg/mL. Following a 12-hour exposure of HIEC-6 cells to sophoridine, samples were collected for subsequent testing.”at Line 84-88.

Q3: L321-323 Unclear as written—rewrite.

Response:Thanks to your suggestions, we've rewritten Line 321-323:” The mRNA expression level of the PDCoV N protein exhibited a significant difference in its inhibition of PI3K when treated with sophoridine compared to the Vector (Fig 7C)..” at Line 320-327.

Q4:L355-357 missing comparative info from ref 22—these authors compared transcriptomic analysis of PDCoV infection in human (HIEC-6) and pig IPEC-J2 intestinal cells. Also, comparisons including ones done on this data for Fig 6 are not discussed.

Response: Thank you very much for your issues raised, We have made the following changes:” In a study conducted by Cruz-Pulido et al., a comparison was made between the transcriptomes of PDCoV-infected HIEC-6 and IPEC-J2 cells, revealing striking similarities in the infection pathways[22]. In the present work, we compared our sequencing results with the 24-hour PDCoV-infected HIEC-6 cell data from Cruz-Pulido et al. However, due to the differential sequencing methodology employed, our study yielded a higher overall factor than that reported by Cruz-Pulido et al.” at Line 359-365.

Q5: L 360 Ref 22 missing and no discussion of comparative findings

Response: Thank you very much for your careful reviews. We have added.

Q6: L 370-372 poorly written sentence-rewrite. For example: Suppression of the PI3K and P38 MAPK signaling pathways through sophoridine in the HIEC-6 cells augmented PDCoV infection.

Response: Thank you for reading this article in detail and asking questions, we have made the following changes to it. ”In the present experimental study, the inhibitory effects of sophoridine on the PI3K and P38 MAPK signaling pathways were investigated about Porcine Deltacoronavirus (PDCoV) infection in HIEC-6 cells. Interestingly, the inhibition of both pathways resulted in a significant increase in PDCoV infection in HIEC-6 cells within a time frame of 12 hours post-inhibition.” at Line 377-382.

Q7:  L374-376 Highlight where the data on KITLG and EFNA1 genes is presented in the results and tie in this data to the discussion. The lengthy discussion of these genes should be germane to the findings in this study instead of a general discussion. 

Response: Thank you very much for your careful reviews. We have deleted and simplified the language in the section you suggested. at Line 386-392.

Figures:

Q1:Fig 1 legend not modified per revised Fig 1—300um marker bar should be 100um and where is this fig?

Response: Thank you very much for your careful reviews. We have corrected it at Line 203.

Q2:Fig. 2. A PDCoV versus mock infection… The quality of Fig 2A is poor and unlikely to reproduce well in the paper.

Response: Thank you for the suggestion, we have made the change.

Q3: Fig.3. Legend is Unclear as written—rewrite.

Response: We appreciate the reviewer’s suggestions and rewrite here。The procedure of transcriptome analysis of PDCoV-infected HIEC-6 cells (Mock is HIEC-6 cells collected at 12h without PDCoV virus infection, PDCoV is HIEC-6 cells collected at 12h, 24h, and 48h, respectively, after PDCoV infection,) at Line 230-232

Q4:Fig.4, 6. VENN—not veen?

Response: Thank you for the suggestion, We have corrected it.

Q5: Fig.7. Legend is not written and does not provide info on the treatments (sophoridine, DMSO)

Response: We have accepted your suggestions and made the changes at Lines 331-332. Protein and qPCR analysis of HIEC-6 cells infected with PDCov after inhibition of the PI3K-Akt & P38 MAPK pathway using sophoridine, And vector is HIEC-6 cells with only DMSO added, sophoridine is HIEC-6 cells with sophoridine added at Line 338-339.

Reviewer 3 Report (Previous Reviewer 3)

Comments and Suggestions for Authors

The manuscript by Jihang et al. analyses new transcriptomic data that describe a significant novel pathway of PDCoV when infecting human cells.

This study also identifies potential cellular proteins as targets for therapeutic drugs.

Although the study is well presented and organised, there are a few comments and corrections that I would like the authors to address.

- In section 2.1 'Cells and Antibodies', in line 86, you have described one of the compounds tested in this study. The title 'Cell, Antibodies and Drugs or Inhibitors' would be more appropriate for this study. Additionally, it may be beneficial to include a section in the 'Materials and Methods' that describes the inhibitors used in the study.

- Also, in this section (line 80), you used the abbreviation 'ST' without explaining its meaning. Before using an abbreviation, it is important to explain its meaning.

- In line 95, a corrected paragraph states that 'after incubation with 10ug trypsin, DMEM was removed'. It would be clearer to state that 'after 2 hours of incubation, DMEM was removed'. Lines 95 and 96 are somewhat confusing. Please revise.

- Please provide an explanation of what IP means in line 113. -

- In lane 136, you stated that you used two flasks as independent biological replicates, but later in Table 2, you listed three replicates. 

 Please correct:

- In the figure legends, please be careful where you place the corrections (highlighted in red in the manuscript). In line 208, you mentioned the scale bar as a part of figure 1A when it should be part of figure 1B. Please make the necessary correction.

- It appears that the same phrase is repeated in lines 241 and 242. Please clarify that section.

-Please, correct Figure legend 5 as it seems the name of the transcript does not correspond to the figure. For example, in the text, Fig 5F refers to IRF when in fact is GAPDH (what we can see in the figure).

-In lane 314 it the authors put "P38 MAPK signalling pathway" and then refer to table S17. However, in table S17 only says MAPK signalling pathway. please correct.

-Table S18 you have a column with gene names so the reader can look for KITLG and EFNA1 as you suggested in line S18.

Thanks a lot!

Author Response

Reviewer 3

Comments to the Author (blind)

The manuscript by Jihang et al. analyses new transcriptomic data that describe a significant novel pathway of PDCoV when infecting human cells.

This study also identifies potential cellular proteins as targets for therapeutic drugs.

Although the study is well presented and organized, there are a few comments and corrections that I would like the authors to address.

Response: Thank you for the positive comment and the constructive suggestions for improving the manuscript. We have revised our manuscript according to all the issues listed below.

Q1:- In section 2.1 'Cells and Antibodies', in line 86, you have described one of the compounds tested in this study. The title 'Cell, Antibodies and Drugs or Inhibitors' would be more appropriate for this study. Additionally, it may be beneficial to include a section in the 'Materials and Methods' that describes the inhibitors used in the study.

Response: Thank you for reading this article in detail and asking questions, we have made the following changes to it. ” Sophoridine was solubilized in dimethyl sulfoxide (DMSO) and subsequently stored. Before usage, completely solubilized sophoridine was introduced into the cell culture medium DMEM at a concentration of 0.2 mg/mL. Following a 12-hour exposure of HIEC-6 cells to sophoridine, samples were collected for subsequent testing..” at Line 84-88.

Q2:- Also, in this section (line 80), you used the abbreviation 'ST' without explaining its meaning. Before using an abbreviation, it is important to explain its meaning.

Response: Thank you very much for your careful reviews. We have added.

Q3:-- In line 95, a corrected paragraph states that 'after incubation with 10ug trypsin, DMEM was removed'. It would be clearer to state that 'after 2 hours of incubation, DMEM was removed'. Lines 95 and 96 are somewhat confusing. Please revise.

Thank you very much for your careful reviews. We reorganized the statements, For PDCoV infection, the virus was diluted using DMEM containing 10 μg/ mL trypsin (Solarbio, Beijing, China; T1350) added to 90% of the ST cells, and the cells were repeatedly freeze-thawed three times until they reached 50% lesion. at Lines 92-95.

Q4:- Please explain what IP means in line 113. –

Response: Thank you very much for your careful reviews. We have added.

Q5:- In lane 136, you stated that you used two flasks as independent biological replicates, but later in Table 2, you listed three replicates.

Response: Thank you very much for your careful reviews. We have corrected it.

 Please correct:

Q6:- In the figure legends, please be careful where you place the corrections (highlighted in red in the manuscript). In line 208, you mentioned the scale bar as a part of Figure 1A when it should be part of Figure 1B. Please make the necessary correction.

Response: Thank you very much for your careful reviews. We have corrected it.

Q7:- It appears that the same phrase is repeated in lines 241 and 242. Please clarify that section.

Response: Thank you for your question we, in this paragraph depicted the data at different times, but the statement was duplicated but the data was at different times and corrected it

Q8:-Please, correct Figure legend 5 as it seems the name of the transcript does not correspond to the figure. For example, in the text, Fig 5F refers to IRF when in fact is GAPDH (what we can see in the figure).

Response: Thank you very much for your careful reviews. We have corrected it.

Q9:--In lane 314 it the authors put "P38 MAPK signalling pathway" and then refer to table S17. However, in table S17 only says MAPK signalling pathway. please correct.

Response: Thank you very much for your careful reviews. We have corrected it.

Q10:--Table S18 you have a column with gene names so the reader can look for KITLG and EFNA1 as you suggested in line S18.

Response: Thank you very much for your careful reviews. We have added.

Round 2

Reviewer 1 Report (Previous Reviewer 1)

Comments and Suggestions for Authors

Comments on the Quality of English Language

Author Response

Point-by-point responses to the comments of the Editor and reviewers

Reviewer 1

Q1:Line 94-95: It is unnecessary to use the word repeatedly when you follow it with the fact that you freeze thawed 3 times. It is redundant. Also what is 50% lesion?

Response: Thank you very much for your careful reviews. We have deleted the word ” repeatedly”. And “lesion” was instead of “cytopathic effect (CPE)” at Lines 94-95.

Q2:Line 95-96: Titer determination by plaque assay and TCID50 are expressed in different units. Plaque assay is expressed as example: 1.23 x 108 pfu/mL. TCID50 would be expressed as 1.23 x108 TCID50/mL. Some clarification is required here. The conversion factor between the two methods is 0.7.

Response: We are so sorry for that. We have corrected this instance to address a misunderstanding resulting from the lack of clarity in our explanation.

Figures:

Q1:Figure 2D: Do not mention the P values for one and two asterisks since they are not represented on your graph. As I mentioned previously you used the student t-test when you had 4 time points for comparison. What comparisons did you do?

Response: We greatly appreciate your inquiry. We have deleted the P values. In Figure 2C and 2D, ANOVA was employed.

Q2: “The Statistics information should be at the end of the caption. Does it describe D as well? You should be using ANOVA, I already said this previously. Please explain why you used the student’s t-test for so many groups.” This is what I said last time. Your comparisons are invalid.

Response: Thank you very much for your careful reviews. We have put the statistics We apologize for the confusion, in Figures 2C & 2D we used ANOVA, but in Figures 7, C, D & E we used t-Test.

Q3:Figure 4B: You have these volcano plots lined up together so they should all have the same scale on the y-axis for visual comparison of any differences. This has not been corrected. I am still seeing VEEN instead of Venn in your caption. Any statistics at all here?

Response: We are so sorry for that. We have corrected.

Q4: Figures 6A and B are Venn Diagrams not veen-still not corrected

Response: We are so sorry for that. We have corrected.

Q5: Figures 7C,D&E: Normalization is misspelled on the y -axis-still not corrected Figure 7 caption. p-values- some of which are not represented in the figure, so are not necessary.

Response: We are so sorry for that. We have corrected the word ”Normalization” and deleted some P-values in the caption.

Reviewer 2 Report (Previous Reviewer 2)

Comments and Suggestions for Authors

Figure 1. Proliferation of PDCoV in ST cells. (A) Process of PDCoV infection in ST cells 208 (B) ST cell lesion caused by PDCoV infection, Red arrow positions are cytopathic lesions. The 209 scale bar corresponds to 1-300 μm 

For Fig 1 and 2 marker bar shown is 100um!

Comments on the Quality of English Language

Improved but still minor errors!

Author Response

Q1:Please, confirm that in Figures 1 and 2 the scale bar (100 µm) is correct

Response: We are so sorry for that. We have corrected the scale bar with 100×.

This manuscript is a resubmission of an earlier submission. The following is a list of the peer review reports and author responses from that submission.

Round 1

Reviewer 1 Report

Comments and Suggestions for Authors

Comments on the Quality of English Language

Reviewer 2 Report

Comments and Suggestions for Authors

The authors conducted a transcriptomic analysis of PDCoV-infected HIEC-6 cells vs uninfected control cells at 3 different time points. From the DEG and KEGG, they identified PI3K-Akt and P38 MAPK as enrichment pathways significantly related to viral infection.

This is largely a descriptive study. Other than characterizing DEG at 3 time points, it is mainly a confirmatory transcriptomic analysis similar to that  described in a prior paper published in Viruses in 2021 (cited as ref 17). This published paper described the detailed transcriptomic analysis of PDCoV-infected human intestinal HIEC-6 cells vs swine IPEC-J2 cells at 24 hr post-infection. Although the authors compared their data at 24 hr post-infection of HIEC-6 cells (Fig 6A, 6B) to that in the published paper (ref 17), they did not address or discuss the large differences seen in Fig 6 A,B between the 2 studies.  Also in Fig 6B, the number of DEGs at 24hr do not match the 1274 cited under 3.3, l 218. The authors should verify these numbers in the text and other Figs as well.

The significance of the study could have been enhanced by investigating the impact of using sophoridine, as in the other studies cited, to downregulate the PI3K-Akt and P38 MAPK  pathways to verify the antiviral effect on PDCoV infection.

Other comments;

1. l 74- The sources for the ST cells and the HIEC-6 cells and the passage numbers should be provided. 

2. L 134- library’s?; l135 QrT-PCR=RT-qPCR? 

3. For comparisons, the same data in Figs 1 and 2 should be shown for PDCoV infection of both ST and HIEC-6 cells, especially since in l 310-312, the morphological changes in the infected HIEC-6 cells are noted but not shown like in Fig 1B for the ST cells.  In addition, it would have been of interest to compare the transcriptomic analysis of PDCoV-infected ST cells to that of the HIEC-6 cells as another control. 

4. The proteins expressed by the KITLG and EFNA1 genes and their function are not defined or discussed (abstract, l287-289, 331,332)

5. What is the human papiliomavirus infection pathway that is upregulated like for PI3K-Akt, but not discussed (Fig 7A)?

6. The text in Figs 4D and 6C is too small to read. Fig 6 C and D appear to be reversed. 

7. In Fig 6 and 7 heat maps, the vertical lanes are unlabeled, but presumably they represent the 3 time points; also which is mock vs infected in Fig 7B? 

Comments on the Quality of English Language

adequate

Reviewer 3 Report

Comments and Suggestions for Authors

Dear authors,

The manuscript of Jiang et al. is very clear, well written and communicates  the research through clear and organised presentation.

Furthermore, the methodology and the results are  well-structured. However, I have some comments that I think it can improve the manuscript:

1.-Regarding the references, I found some publications online, related to the manuscript, that are not cited. Some examples are the following:

https://www.sciencedirect.com/science/article/pii/S0161589021000894?casa_token=aoHKyGqk4bsAAAAA:R0halHCTXEkTV9Grucwct9OMgbBNuf6twEDhQyrbCMM6OJ2pgo6njgQ0lHZ25YqyMqkfdrep

https://journals.plos.org/plosone/article?id=10.1371/journal.pone.0223177

Adding more references to the discussion section will improve it.

2.-Most of figures are clear but some details must be taken into account.

In figure 3, "experimental process"; in the figure leyend you talk about transcriptome analysis however, in the figure you put "LC-MS/MS and refer o proteomics. Can you correct or explain this?

3. You should add a table or list of the up regulated and down regulated genes for each sample (or at least the most important). This should be add as a supplemental material. This will help the reader to better understand/analyse and compare your results with others.

Thank you very much.

Kind regards,